# Real-World Outcomes of Patients with Advanced Epidermal Growth Factor Receptor-Mutated Non-Small Cell Lung Cancer in Canada Using Data Extracted by Large Language Model-Based Artificial Intelligence

Ruth Moulson [1], Jennifer Law [2], Adrian Sacher [2], Geoffrey Liu [2], Frances A. Shepherd [2], Penelope Bradbury [2], Lawson Eng [2], Sandra Iczkovitz [3], Erica Abbie [3], Julia Elia-Pacitti [3], Emmanuel M. Ewara [3], Viktoriia Mokriak [1], Jessica Weiss [1], Christopher Pettengell [1] and Natasha B. Leighl [2,*]

1   Pentavere, 460 College Street, Toronto, ON M6G 1A1, Canada; rmoulson@pentavere.com (R.M.)
2   Department of Medical Oncology, Princess Margaret Cancer Centre, University Health Network, Toronto, ON M5G 2C1, Canada
3   Janssen Inc., Toronto, ON M3C 1L9, Canada
*   Correspondence: natasha.leighl@uhn.ca

**Abstract:** Real-world evidence for patients with advanced *EGFR*-mutated non-small cell lung cancer (NSCLC) in Canada is limited. This study's objective was to use previously validated DARWEN[TM] artificial intelligence (AI) to extract data from electronic heath records of patients with non-squamous NSCLC at University Health Network (UHN) to describe *EGFR* mutation prevalence, treatment patterns, and outcomes. Of 2154 patients with NSCLC, 613 had advanced disease. Of these, 136 (22%) had common sensitizing *EGFR* mutations (c*EGFR*m; ex19del, L858R), 8 (1%) had exon 20 insertions (ex20ins), and 338 (55%) had *EGFR* wild type. One-year overall survival (OS) (95% CI) for patients with c*EGFR*m, ex20ins, and *EGFR* wild type tumours was 88% (83, 94), 100% (100, 100), and 59% (53, 65), respectively. In total, 38% patients with ex20ins received experimental ex20ins targeting treatment as their first-line therapy. A total of 57 patients (36%) with c*EGFR*m received osimertinib as their first-line treatment, and 61 (39%) received it as their second-line treatment. One-year OS (95% CI) following the discontinuation of osimertinib was 35% (17, 75) post-first-line and 20% (9, 44) post-second-line. In this real-world AI-generated dataset, survival post-osimertinib was poor in patients with c*EGFR* mutations. Patients with ex20ins in this cohort had improved outcomes, possibly due to ex20ins targeting treatment, highlighting the need for more effective treatments for patients with advanced *EGFR*m NSCLC.

**Keywords:** real-world evidence; artificial intelligence; non-small cell lung cancer

## 1. Introduction

Lung cancer is the most common cancer diagnosis in Canada, with an estimated 1 in 15 Canadians receiving a diagnosis in their lifetime [1]. While the prognosis and outcomes of lung cancer have improved in recent decades, largely as a result of novel, innovative therapies and increased awareness of the risk factors, this disease remains the deadliest cancer in Canada [1,2]. Approximately 85% of patients with lung cancer present with NSCLC, with up to two-thirds harbouring actionable driver mutations, most commonly occurring in the *epidermal growth factor receptor* (*EGFR*) [3–5]. *EGFR* mutations can be categorized based on the type of mutation and the exon in which they occur. Exon 19 deletions (ex19del) and exon 21 L858R point mutations account for up to 90% of all *EGFR* mutations and are often referred to as common sensitizing *EGFR* mutations (c*EGFR*m) [6]. The third most frequently occurring mutations are exon 20 insertion mutations (ex20ins) and represent approximately 1–12% of all *EGFR* mutations, and 0.1–4% of all NSCLC

mutations [7]. However, uncertainty in the real-world estimates of these mutations exist, partly due to the evolution of testing methods, with recent guidelines recommending next-generation sequencing (NGS) for identifying actionable driver alterations, such as *EGFR* [8,9]. This technique has improved sensitivity, can detect mutations using a smaller amount of DNA, and sequences a greater part of the gene compared with the historical standard, polymerase chain reaction (PCR), which is limited to specific loci and can miss up to 50% of ex20ins mutations, but it requires a smaller tissue sample than NGS [10–12].

The treatment of patients with *EGFR* mutations has been revolutionized by tyrosine kinase inhibitor (TKI) targeted therapy. The recommended first-line therapy for advanced-stage patients with c*EGFR*m in Canada is the third-generation kinase inhibitor, osimertinib [13,14]. However, the long-term benefit of this therapy is limited by the development of acquired resistance via multiple mechanisms [15]. Recently, multiple new options for overcoming osimertinib resistance have emerged, including amivantamab + lazertinib, chemotherapy, local therapy (surgery or radiation), chemotherapy + amivantamab/lazertinib, antibody-drug conjugates (ADCs), including patritumab deruxtecan and datopotamab deruxtecan, and combined targeted therapies against emergent targetable alterations (e.g., for MET amplification: osimertinib + savolitinib and tepotinib + osimertinib) [16]. These emerging treatment options are particularly important as many patients with c*EGFR*m who are treated with a first-line TKI die before receiving a second-line one [17]; thus, there remains a high unmet need for effective and safe therapies early in patients' treatment journeys, and there is currently a lack of real-world evidence (RWE), specifically in the Canadian setting, on patients with c*EGFR*m who may benefit from these therapies.

Independent of acquired resistance, ex20ins are associated with limited response to TKIs [18]. Compared with other *EGFR* mutations, patients with ex20ins have especially poor prognosis, with markedly reduced sensitivity to approved *EGFR* kinase inhibitors [18–20]. Until recently, there have been limited treatment options for patients with ex20ins, with the recommended first-line treatment being either platinum-based chemotherapy or clinical trial [13]. However, the Canadian treatment landscape is evolving, as the results from the phase III PAPILLON study have established amivantamab + chemotherapy as a new first-line standard for this patient population [21]. As the treatment landscape changes, there is a need to gain a better understanding of the patients who may benefit from these newer therapies.

Over the past two decades, the generation of RWE from electronic health record (EHR) systems has contributed new insights into the prevalence of lung cancer subtypes and the disease characteristics and clinical outcomes for these patients. Through the routine collection of clinical evidence, real-world data (RWD) from EHRs can be harnessed to study disease progression, treatment patterns, and measure survival outcomes over time. Recent advances in artificial intelligence (AI) and Natural Language Processing (NLP) have enabled the extraction and analysis of RWD from clinical documentation and unstructured text (such as clinical notes and lab results) housed within EHR systems, with higher accuracy and at a significantly greater scale than manual abstraction, the current standard practice for extracting RWD from EHRs [22,23]. It is increasingly being recognized that these technologies play an important role in clinical medicine by allowing clinician's and researchers access to previously inaccessible data, which can be used to inform clinical decision making and enhance clinical care [24].

The aim of this study was to leverage the previously validated, commercially available AI technology, Pentavere's DARWEN^TM, to identify patients and extract RWD from EHRs at the University Health Network Princess Margaret Cancer Centre (UHN-PMCC), the largest cancer-treating centre in Canada, to understand the prevalence, treatment patterns, and clinical outcomes of patients diagnosed with advanced c*EGFR*m (ex19del and exon 21 L858R) and ex20ins mutations.

## 2. Materials and Methods

### 2.1. Study Design

This was a retrospective cohort study of data elements from EHRs stored at the UHN-PMCC using AI technology. The AI engine combines large language models and an ensemble of other techniques that have previously been evaluated and validated against manual abstraction across multiple disease domains, including lung cancer [22,25], breast cancer [26], dermatology [27], and infectious diseases [28] at multiple Canadian institutions, including the UHN-PMCC.

The study period extended from 1 January 2017 to 1 March 2022 and used the institutional Cancer Registry. All adult patients who were ≥18 years of age with non-squamous NSCLC and seen at the UHN-PMCC during the study period were included in the study. Follow-up data from EHRs were included up to the extent that they were available within the study period. The initial list of patients was provided from the UHN-PMCC's Molecular Testing Database.

### 2.2. Data Extraction

Clinical features extracted included mutation status, clinical and demographic characteristics, treatment information, and clinical outcomes. Data were extracted directly from the EHRs of all patients with non-squamous NSCLC seen at the UHN-PMCC between 1 January 2017 and 1 March 2022. The AI engine was installed on the UHN-PMCC's infrastructure and used to extract relevant data variables directly from the source systems where available. Clinical outcomes were derived using the extracted data, including time to treatment discontinuation (TTD) and overall survival (OS). All features were extracted following a set of pre-defined rules and definitions developed by the UHN-PMCC Principal Investigator. DARWEN$^{TM}$ AI has previously been validated against the manual chart review for the same clinical features at the UHN-PMCC, the process for which has previously been described [22].

### 2.3. Outcomes

The primary outcome of interest was mutation prevalence. Other outcomes of interest included the frequency of patients receiving each type of therapy by line of therapy (LoT), time from diagnosis to treatment initiation per LoT, and clinical outcomes, including TTD, OS, and OS post-osimertinib. TTD was measured from the date of the treatment initiation of one line of therapy to the last known date of the treatment of the same line of therapy. TTD was derived for first-line, second-line, and third-line therapies. OS was measured from date of diagnosis to date of death, and from date of treatment initiation to date of death for first-line and second-line therapies. Patients who did not experience the event before the study's end period were censored at their date of last follow-up or the study's end date, whichever came first. Overall survival, specifically for patients who had discontinued osimertinib, was explored and measured from the stop date of osimertinib to date of death. Patients who did not experience the event before the study's end period were censored at their date of last follow-up or the study's end date, whichever came first. OS was derived from the end of first-line osimertinib and the end of second-line osimertinib.

### 2.4. Statistical Analyses

Descriptive analyses were performed to summarize the patients' demographics, disease characteristics, treatment patterns, and outcomes of interest across the study cohort. Continuous variables were described using mean and standard deviation (SD) and the median and range. Categorical variables were described by frequencies and related percentages. The number of missing observations was reported for all variables. Time to event(s) was described using Kaplan–Meier curves that visually estimated the distribution of times to some events (e.g., OS) and accounted for patients for whom the event had not yet occurred, i.e., following standard censoring rules. Numbers at risk and the cumulative number of events were reported for each curve.

## 3. Results

### 3.1. Patients

Between 1 January 2017 and 1 March 2022, 2154 patients were identified with non-squamous NSCLC and were seen at the UHN-PMCC. Of these patients, 613 patients had advanced-stage disease, of which 136 (22%) patients had c*EGFR*m at diagnosis, 8 (1%) had ex20ins at diagnosis, 338 (55%) had *EGFR* wild type tumours at diagnosis, and 131 (21%) did not have mutation testing at diagnosis conducted at the UHN-PMCC. A flow diagram of the included patients is presented in Supplementary Figure S1.

Across all 613 patients with advanced-stage disease, median (range) age at advanced diagnosis was 67 years (27–96); 51% of patients were male, 84% had adenocarcinoma, and 38% had never been smokers. At advanced diagnosis, 30% of patients presented with bone metastases, and 14% had brain metastases (Table 1). The majority of patients (81%) were diagnosed at the UHN-PMCC. Of the 131 patients who did not have mutation testing at the UHN-PMCC, 56% were also not diagnosed at the UHN-PMCC, and all 131 were not included in the clinical outcome analyses. The median (range) duration of the follow-up from diagnosis for all patients was 12.3 months (0.0–61.8) (Table 1). AI validation metrics for the AI-extracted clinical features are presented in Supplementary Table S1.

**Table 1.** Clinical, demographic, and disease characteristics of advanced-stage NSCLC patients stratified by EGFR mutation status at diagnosis.

| | Common Sensitizing *EGFR* (N = 136) | *EGFR* Wild Type [b] (N = 338) | Exon 20 Insertion (N = 8) | *EGFR* Test Not Conducted at UHN (N = 131) | Total (N = 613) |
|---|---|---|---|---|---|
| **Age at diagnosis** | | | | | |
| Mean (SD) | 65.1 (11.6) | 67.6 (11.6) | 59.9 (19.3) | 65.3 (10.6) | 66.5 (11.6) |
| Median (range) | 65.0 (34.0, 91.0) | 68.0 (27.0, 96.0) | 59.0 (38.0, 88.0) | 66.0 (32.0, 88.0) | 67.0 (27.0, 96.0) |
| **Sex** | | | | | |
| Female | 89 (65.4%) | 146 (43.2%) | 4 (50.0%) | 64 (48.9%) | 303 (49.4%) |
| Male | 47 (34.6%) | 192 (56.8%) | 4 (50.0%) | 67 (51.1%) | 310 (50.6%) |
| **Histology** | | | | | |
| Adenocarcinoma | 129 (94.9%) | 276 (81.7%) | 7 (87.5%) | 102 (77.9%) | 514 (83.8%) |
| Adenosquamous | 0 (0.0%) | 2 (0.6%) | 1 (12.5%) | 0 (0.0%) | 3 (0.5%) |
| Large cell | 2 (1.5%) | 21 (6.2%) | 0 (0.0%) | 16 (12.2%) | 39 (6.4%) |
| Sarcomatoid | 0 (0.0%) | 5 (1.5%) | 0 (0.0%) | 6 (4.6%) | 11 (1.8%) |
| Non-small cell (unspecified) | 5 (3.7%) | 34 (10.1%) | 0 (0.0%) | 7 (5.3%) | 46 (7.5%) |
| **Smoking status** | | | | | |
| Smoker | 8 (5.9%) | 101 (29.9%) | 0 (0.0%) | 30 (22.9%) | 139 (22.7%) |
| Former smoker | 29 (21.3%) | 152 (45.0%) | 2 (25.0%) | 50 (38.2%) | 233 (38.0%) |
| Never smoked | 98 (72.1%) | 83 (24.6%) | 6 (75.0%) | 48 (36.6%) | 235 (38.3%) |
| Missing | 1 (0.7%) | 2 (0.6%) | 0 (0.0%) | 3 (2.3%) | 6 (1.0%) |
| **Weight Category** | | | | | |
| <80 kg | 105 (77.2%) | 241 (71.3%) | 6 (75.0%) | 98 (74.8%) | 450 (73.4%) |
| ≥80 kg | 17 (12.5%) | 59 (17.5%) | 0 (0.0%) | 21 (16.0%) | 97 (15.8%) |
| Missing | 14 (10.3%) | 38 (11.2%) | 2 (25.0%) | 12 (9.2%) | 66 (10.8%) |

**Table 1.** *Cont.*

| | Common Sensitizing *EGFR* (N = 136) | *EGFR* Wild Type [b] (N = 338) | Exon 20 Insertion (N = 8) | *EGFR* Test Not Conducted at UHN (N = 131) | Total (N = 613) |
|---|---|---|---|---|---|
| **ECOG at diagnosis** | | | | | |
| 0 | 23 (16.9%) | 34 (10.1%) | 1 (12.5%) | 17 (13.0%) | 75 (12.2%) |
| 1 | 84 (61.8%) | 180 (53.3%) | 5 (62.5%) | 64 (48.9%) | 333 (54.3%) |
| 2 | 15 (11.0%) | 59 (17.5%) | 1 (12.5%) | 20 (15.3%) | 95 (15.5%) |
| 3 | 9 (6.6%) | 28 (8.3%) | 1 (12.5%) | 13 (9.9%) | 51 (8.3%) |
| 4 | 1 (0.7%) | 5 (1.5%) | 0 (0.0%) | 2 (1.5%) | 8 (1.3%) |
| Missing | 4 (2.9%) | 32 (9.5%) | 0 (0.0%) | 15 (11.5%) | 51 (8.3%) |
| **Organ level metastatic sites at diagnosis** [a] | | | | | |
| Bone | 46 (33.8%) | 119 (35.2%) | 3 (37.5%) | 18 (13.7%) | 186 (30.3%) |
| Brain | 21 (15.4%) | 46 (13.6%) | 1 (12.5%) | 16 (12.2%) | 84 (13.7%) |
| Lung | 22 (16.2%) | 55 (16.3%) | 0 (0.0%) | 14 (10.7%) | 91 (14.8%) |
| Liver | 19 (14.0%) | 42 (12.4%) | 2 (25.0%) | 17 (13.0%) | 80 (13.1%) |
| **Diagnosed at UHN** | | | | | |
| True | 120 (88.2%) | 314 (92.9%) | 5 (62.5%) | 58 (44.3%) | 497 (81.1%) |
| False | 16 (11.8%) | 24 (7.1%) | 3 (37.5%) | 73 (55.7%) | 116 (18.9%) |
| **Follow-up time since diagnosis (months)** | | | | | |
| Mean (SD) | 21.4 (14.1) | 13.6 (14.0) | 24.0 (19.8) | 19.0 (16.0) | 16.6 (14.9) |
| Median (range) | 19.2 (0.4, 58.9) | 8.0 (0.3, 59.4) | 20.2 (0.4, 61.0) | 14.7 (0.0, 61.8) | 12.3 (0.0, 61.8) |

[a] Patients could have had multiple metastatic sites at diagnosis, and therefore percentages may not add up to 100%. Further, patients may have had metastases to body parts other than the bone, brain, lung, and liver, which also explains why percentages may not add up to 100%. [b] Includes patients with a negative *EGFR* test within 3 months of NSCLC diagnosis but does not exclude the possibility of other mutations. ECOG: Eastern Cooperative Oncology Group; NSCLC: non-small-cell lung cancer; SD: standard deviation; UHN: University Health Network.

### 3.2. Treatment Patterns

Treatment patterns were assessed from the date of diagnosis until date of death, date of last follow-up, or the end of the study period, whichever came first. For advanced-stage patients with c*EGFR*m at diagnosis, 129/136 (95%) received first-line therapy, of which 124/129 (96%) received an *EGFR* TKI in their first-line treatment (Figure 1A; Supplementary Table S2). Of patients with c*EGFR*m, 62/136 (46%) did not go on to receive second-line treatment during the study period (Figure 1A) (34 of which received osimertinib in their first-line therapy and 19 of which received gefitinib in their first-line therapy), and 21/62 (34%) of these patients died. Of patients who did go on to receive second-line (74/136 [54%]) and third-line therapies (27/136 [20%]), the most common treatment type was also *EGFR* TKIs in those lines (Figure 1A). Between 2017 and 2019, gefitinib was the most common first-line *EGFR* TKI administered for patients with c*EGFR*m, with 81% of patients who initiated an *EGFR* TKI in 2017–2019 receiving gefitinib (Table 2). Coincident with provincial funding as of January 2020, osimertinib was the most frequently used first-line *EGFR* TKI from 2020 to 2022, with 93% of patients who initiated an *EGFR* TKI in this period receiving osimertinib (Table 2).

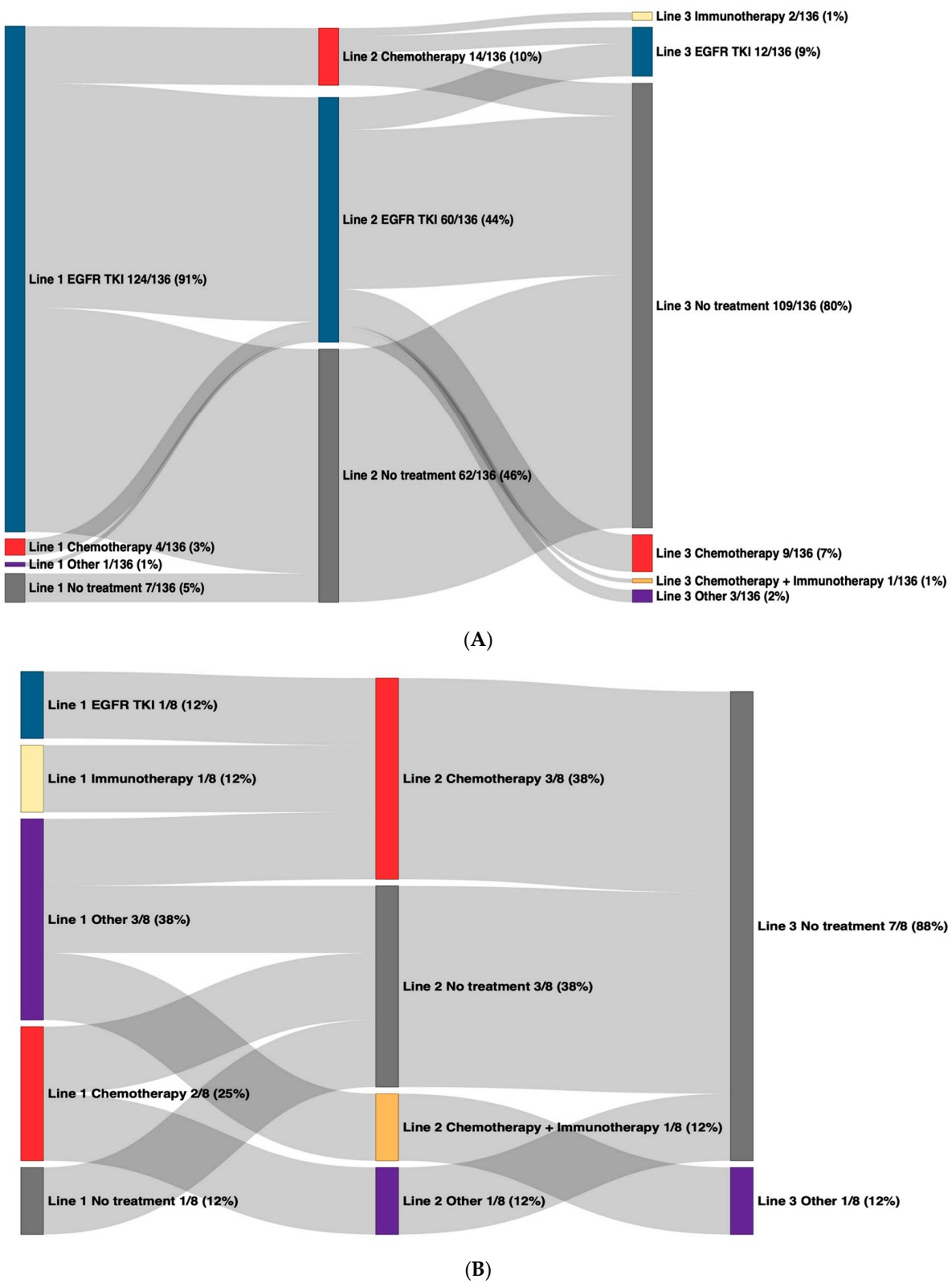

**Figure 1.** Overall treatment patterns in advanced-stage NSCLC patients by mutation status at diagnosis. Line of therapy is denoted by the number followed by the treatment regimen, with first-line on the left and subsequent lines to the right. "Other" includes capmatinib, savolitinib, poziotinib, mobocertinib, lazertinib, and telisotuzumab. *EGFR: epidermal growth factor receptor*; NSCLC: non-small-cell lung cancer; TKI: tyrosine kinase inhibitor. (**A**) Common sensitizing *EGFR* mutations. (**B**) Exon 20 insertion mutations.

For advanced-stage patients with ex20ins at diagnosis, 7/8 (88%) received first-line therapy (Figure 1B), and 3/8 (38%) received the experimental ex20ins targeting TKI, poziotinib (Supplementary Table S1). Second-line therapy was received by 5/8 (63%) patients (4/8 received chemotherapy), and 1/8 (13%) went on to receive third-line therapy (Figure 1B). For advanced-stage patients with *EGFR* wild type tumours at diagnosis, treatment patterns were generally heterogeneous across all lines of therapy (Supplementary Table S1).

**Table 2.** First-line EGFR TKI treatment patterns in advanced-stage NSCLC patients stratified by year of initiating treatment and mutation status at diagnosis.

| | **Common Sensitizing *EGFR*** | | | | | | ***EGFR* Wild Type** | | | **Exon 20 Insertion** |
|---|---|---|---|---|---|---|---|---|---|---|
| | **2017 (N = 17)** | **2018 (N = 36)** | **2019 (N = 28)** | **2020 (N = 17)** | **2021 (N = 24)** | **2022 (N = 2)** | **2017 (N = 1)** | **2019 (N = 3)** | **2021 (N = 5)** | **2018 (N = 1)** |
| Afatinib | 1 (5.9%) | 6 (16.7%) | 2 (7.1%) | 1 (5.9%) | 0 (0.0%) | 0 (0.0%) | 0 (0.0%) | 1 (33.3%) | 4 (80.0%) | 1 (100.0%) |
| Erlotinib | 1 (5.9%) | 0 (0.0%) | 0 (0.0%) | 1 (5.9%) | 0 (0.0%) | 0 (0.0%) | 0 (0.0%) | 0 (0.0%) | 0 (0.0%) | 0 (0.0%) |
| Gefitinib | 15 (88.2%) | 30 (83.3%) | 21 (75.0%) | 1 (5.9%) | 0 (0.0%) | 0 (0.0%) | 1 (100.0%) | 2 (66.7%) | 0 (0.0%) | 0 (0.0%) |
| Osimertinib | 0 (0.0%) | 0 (0.0%) | 5 (17.9%) | 14 (82.4%) | 24 (100.0%) | 2 (100.0%) | 0 (0.0%) | 0 (0.0%) | 1 (20.0%) | 0 (0.0%) |

Bolded N includes patients who initiated a first-line *EGFR* TKI in the specified year. *EGFR: epidermal growth factor receptor*; NSCLC: non-small-cell lung cancer; TKI: tyrosine kinase inhibitor.

The median time from advanced diagnosis to first-line treatment initiation for patients with c*EGFR*m, ex20ins, and *EGFR* wild type tumours was 0.8 months, 2.5 months, and 1.5 months, respectively (Supplementary Table S1). Longer time from advanced diagnosis to first-line treatment initiation was observed for patients with ex20ins, likely due to a lack of clear treatment options for these patients, and time required for clinical trial enrolment.

### 3.3. Clinical Outcomes

For patients with c*EGFR*m, median TTD1 (95% CI), TTD2, and TTD3 was 9.0 months (7.0, 10.3), 6.7 months (4.8, 10.7), and 2.9 months (1.6, 6.8), respectively (Table 3, Figure 2). For patients with ex20ins median, TTD1 (95% CI) and TTD2 were 5.0 months (3.5, NA) and 7.9 months (5.7, NA), respectively (Table 3, Figure 2). For patients with *EGFR* wild type tumours, treatment duration was generally shorter, with median TTD1 (95% CI), TTD2, and TTD3 of 4.0 months (3.3, 4.6), 2.8 months (1.9, 4.8), and 2.1 months (1.3, 5.8), respectively (Table 3, Figure 2).

**Table 3.** Time to event analyses for patients stratified by mutation status at diagnosis.

| Clinical Outcome | 12 Months (95% CI) | 24 Months (95% CI) | Median (95% CI) |
|---|---|---|---|
| | | TTD1 | |
| Exon 20 insertion | 14% (2, 88) | 14% (2, 88) | 5 months (3.5, NA) |
| Common sensitizing *EGFR* | 34% (27, 43) | 12% (7, 19) | 9 months (7, 10.3) |
| *EGFR* wild type | 20% (15, 26) | 7% (4, 11) | 4 months (3.3, 4.6) |
| | | TTD2 | |
| Exon 20 insertion | NA | NA | 7.9 months (5.7, NA) |
| Common sensitizing *EGFR* | 34% (25, 46) | 8% (4, 17) | 6.7 months (4.8, 10.7) |
| *EGFR* wild type | 15% (10, 24) | 4% (1, 10) | 2.8 months (1.9, 4.8) |

**Table 3.** *Cont.*

| Clinical Outcome | 12 Months (95% CI) | 24 Months (95% CI) | Median (95% CI) |
|---|---|---|---|
| TTD3 | | | |
| Exon 20 insertion | NA | NA | 2.8 months (NA, NA) |
| Common sensitizing *EGFR* | 11% (4, 32) | 4% (1, 25) | 2.9 months (1.6, 6.8) |
| *EGFR* wild type | 11% (4, 27) | 3% (0, 19) | 2.1 months (1.3, 5.8) |
| OS from diagnosis | | | |
| Exon 20 insertion | 100% (100, 100) | 80% (52, 100) | NA months (32.1, NA) |
| Common sensitizing *EGFR* | 88% (83, 94) | 63% (54, 73) | 30.1 months (25.2, 38.9) |
| *EGFR* wild type | 59% (53, 65) | 38% (32, 44) | 16.2 months (13.2, 20.5) |
| OS from first-line | | | |
| Exon 20 insertion | 100% (100, 100) | 60% (29, 100) | NA months (18.4, NA) |
| Common sensitizing *EGFR* | 85% (78, 92) | 57% (48, 69) | 26.4 months (23.2, 36.8) |
| *EGFR* wild type | 62% (56, 69) | 38% (32, 46) | 19.3 months (14.2, 22.6) |
| OS from second-line | | | |
| Exon 20 insertion | 75% (43, 100) | NA | 13.1 months (11, NA) |
| Common sensitizing *EGFR* | 56% (45, 69) | 42% (31, 58) | 20.3 months (11, 40.2) |
| *EGFR* wild type | 48% (39, 59) | 26% (17, 38) | 10.6 months (7.6, 15.3) |
| OS from end of first-line osimertinib | | | |
| Common sensitizing *EGFR* | 35% (17, 75) | - | 5.6 months (3.2, NA) |
| OS from end of second-line osimertinib | | | |
| Common sensitizing *EGFR* | 20% (9, 44) | - | 3.3 months (2, 10.4) |

CI: confidence interval; *EGFR: epidermal growth factor receptor*; NA: Not applicable either due to small sample size or confidence interval not reached; OS: overall survival; TTD: time to treatment discontinuation.

The 1-year OS from diagnosis for patients with cEGFRm, ex20ins, and *EGFR* wild type was 88% (83, 94), 100% (100, 100), and 59% (53, 65), respectively (Table 3). OS from first-line and second-line therapies can be found in Table 3.

Of advanced-stage patients with c*EGFR*m, 57 (36%) received first-line osimertinib, and 61 (39%) received second-line osimertinib. After discontinuing osimertinib treatment, OS was low: 1-year OS (95% CI) was 35% (17, 75) post-first-line osimertinib and 20% (9, 44) post-second-line. Median OS was 5.6 months (3.2, NA) post-first-line osimertinib and 3.3 months (2.0, 10.4) post-second-line (Table 3, Figure 3).

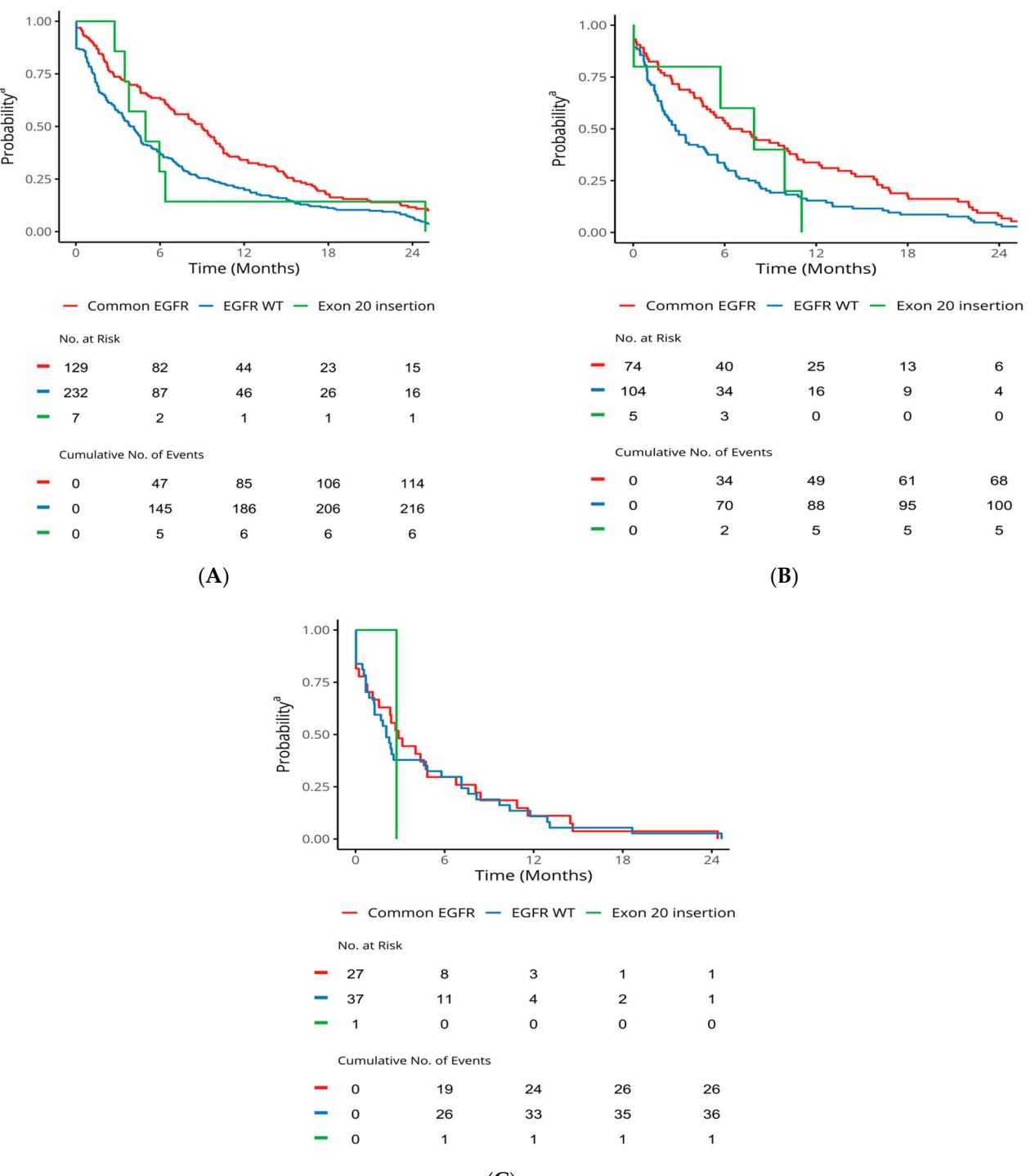

**Figure 2.** TTD in advanced-stage NSCLC patients stratified by mutation status. [a] Probability of staying on the line treatment. *EGFR: epidermal growth factor receptor*; NSCLC: non-small-cell lung cancer; TTD: time to treatment discontinuation. (**A**) TTD1. (**B**) TTD2. (**C**) TTD3.

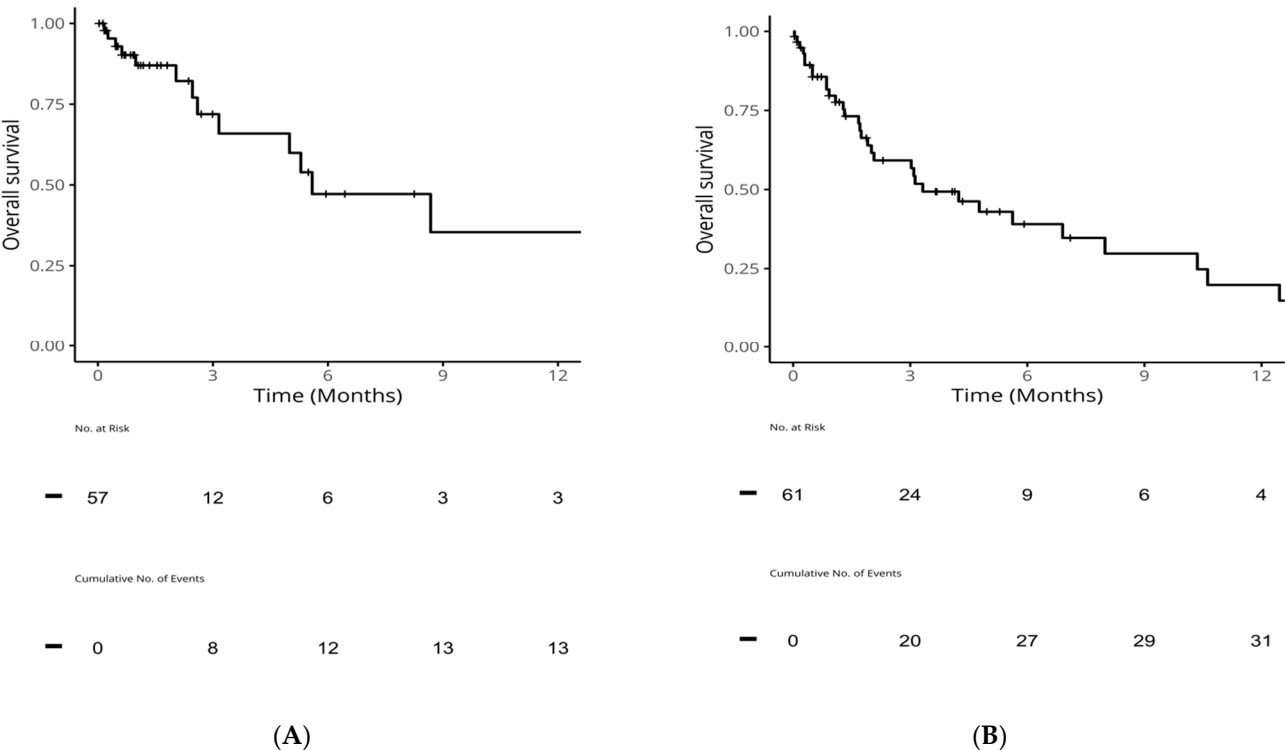

**(A)**  **(B)**

**Figure 3.** OS from end of first-line or second-line in patients with common sensitizing *EGFR* mutations who received osimertinib. (**A**): OS from end of first-line osimertinib; (**B**): OS from end of second-line osimertinib. *EGFR: epidermal growth factor receptor*; OS: overall survival.

### 4. Discussion

This study identified Canadian patients with non-squamous NSCLC at the largest cancer treatment centre in Canada and described the real-world characteristics, treatment patterns, and clinical outcomes for patients with advanced ex19del, exon 21 L858R, and ex20ins *EGFR* mutations using AI-extracted data. It was found that, as expected, patients with c*EGFR*m were primarily treated with *EGFR* TKIs. TKI treatment use changed over time with the approval of novel therapies. From 2020, osimertinib emerged as the most frequently administered *EGFR* TKI, in line with the treatment guidelines. Importantly, it was found that patients with c*EGFR*m treated with osimertinib progressed on therapy and exhibited poor survival rates after discontinuing treatment, emphasizing the need for more efficacious therapies earlier in patients' treatment journeys. It was also found that several patients with ex20ins were treated with the experimental ex20ins TKI, poziotinib, and may have had better survival as a result.

Among 2154 patients with non-squamous NSCLC and seen at the UHN-PMCC during the study period, 613 had advanced disease, of which 1% had ex20ins at diagnosis, consistent with other real-world estimates in Canada, and at the UHN-PMCC [29–31], median time from advanced diagnosis to initiating first-line therapy was longer for patients with ex20ins in comparison to patients with c*EGFR*m (2.5 months versus 0.8 months, respectively), likely due to the absence of a clear first-line targeted treatment option for these patients, coupled with the time required for clinical trial enrolment.

A recent European RWE registry study investigated the use of different treatment types and their impact on survival rates among patients with EGFR ex20ins mutations. Novel targeted agents, including amivantamab, mobocertinib, and poziotinib, were associated with improved survival rates in the first-line setting. As well, in the multivariate analysis, type of treatment (novel targeted therapy versus chemotherapy) had a significant effect on OS ($p$ = 0.03) [32]. In this study, of patients with ex20ins, 38% received the experimental exon 20 targeting TKI, poziotinib, in their first-line therapy and achieved better survival

than patients with c*EGFR*m or *EGFR* wild type, emphasizing the benefit of novel, targeted therapies; although, it is important to acknowledge the limitation of the survival analyses for the ex20ins patient group in this study due to the small sample size associated with this rare mutation. However, in the phase II trial of poziotinib, serious adverse events were observed, including grade $\geq 3$ diarrhoea and rash, leading to treatment interruptions, which could explain the shorter TTD1 for patients with ex20ins in this study compared with c*EGFR*m. Further, the recent phase III trial of mobocertinib in first-line therapy for ex20ins patients was terminated early due to futility. These results highlight the need for efficacious and safe exon 20 targeting therapies to improve survival outcomes for these patients, in alignment with the evolving treatment landscape.

Over the study period, treatment patterns for patients with c*EGFR*m evolved with the introduction of novel third-generation *EGFR* TKIs. From 2017 to 2019, gefitinib was the predominant first-line *EGFR* TKI, followed by osimertinib in 2020–2022. However, it is noteworthy that 62/136 (46%) of patients with c*EGFR*m (34 of which received osimertinib in their first-line treatment) did not go on to receive second-line therapy during the study period, and of these patients, 21/62 (34%) died. For patients with c*EGFR*m who received osimertinib either in their first-line or second-line therapies, OS following the discontinuation of osimertinib was poor (1-year OS [95% CI] was 35% (17, 75) post-first-line osimertinib), aligning with findings observed in the RWE study of US databases conducted by Girard et al. (2023) [33]. These observations highlight the importance of effective novel treatment options early in patients' treatment journeys. Further studies may wish to investigate the specific risk factors associated with the mortality of patients prior to receiving second-line therapy.

As this study was a retrospective study of data extracted from EHRs, limitations due to the availability and accuracy of data captured in the EHR were observed. For example, many patient deaths occurred in the community setting rather than the hospital, and dates of death are only collected when hospitals are notified of a patient's death, which may have resulted in missing mortality data. This could have led to higher levels of data censoring in Kaplan–Meier curves and survival analyses. Additionally, at the UHN-PMCC, oral therapy prescription data are only dictated into the clinical notes and, therefore, these records are susceptible to incompleteness and human dictation error. Further, as this study was conducted at one urban treatment site in Toronto, Ontario, the cohort may not accurately represent the wider provincial or national population and may not be directly reproducible; however, the prevalence rates observed in this study do align with previous studies in Canada and at the UHN-PMCC [29–31].

## 5. Conclusions

This study identified patients with non-squamous NSCLC at one of Canada's largest cancer treatment centres using previously validated AI technology. Using these types of technologies allows for the extraction of previously unavailable data in a more consistent, efficient, and scalable way compared to manual chart review [22]. The results from this study highlight the importance of effective novel targeted therapies for improving survival outcomes in patients with ex20ins *EGFR* mutations, in alignment with the evolving treatment landscape for first-line therapy. The findings also emphasize the need for optimal therapies early in the treatment of patients with c*EGFR*m.

**Supplementary Materials:** The following supporting information can be downloaded at: https://www.mdpi.com/article/10.3390/curroncol31040146/s1, Table S1: Evaluation of DARWEN$^{TM}$ AI. Table S2: First-line treatments in advanced-stage NSCLC patients stratified by mutation status at diagnosis. Figure S1: Summary of included patients.

**Author Contributions:** R.M.: Conceptualization; Data Curation; Formal Analysis; Funding Acquisition; Investigation; Methodology; Project Administration; Resources; Software; Supervision; Validation; Visualization; Roles/Writing—Original Draft; Writing—Review and Editing. J.L.: Conceptualization; Data Curation; Formal Analysis; Funding Acquisition; Investigation; Methodology;

Project Administration; Resources; Supervision; Validation; Visualization; Roles/Writing—Original Draft; Writing—Review and Editing. A.S.: Conceptualization; Data Curation; Funding Acquisition; Investigation; Methodology; Resources; Validation; Roles/Writing—Original Draft; Writing—Review and Editing. G.L.: Conceptualization; Data Curation; Funding Acquisition; Investigation; Methodology; Resources; Validation; Roles/Writing—Original Draft; Writing—Review and Editing. F.A.S.: Conceptualization; Data Curation; Funding Acquisition; Investigation; Methodology; Resources; Validation; Roles/Writing—Original Draft; Writing—Review and Editing. P.B.: Conceptualization; Data Curation; Funding Acquisition; Investigation; Methodology; Resources; Validation; Roles/Writing—Original Draft; Writing—Review and Editing. L.E.: Conceptualization; Data Curation; Funding Acquisition; Investigation; Methodology; Resources; Validation; Roles/Writing—Original Draft; Writing—Review and Editing. S.I.: Conceptualization; Formal Analysis; Funding Acquisition; Investigation; Methodology; Project Administration; Resources; Supervision; Validation; Visualization; Roles/Writing—Original Draft; Writing—Review and Editing. E.A.: Conceptualization; Formal Analysis; Funding Acquisition; Investigation; Methodology; Project Administration; Resources; Supervision; Validation; Visualization; Roles/Writing—Original Draft; Writing—Review and Editing. J.E.-P.: Conceptualization; Formal Analysis; Funding Acquisition; Investigation; Methodology; Project Administration; Resources; Supervision; Validation; Visualization; Roles/Writing—Original Draft; Writing—Review and Editing. E.M.E.: Conceptualization; Formal Analysis; Funding Acquisition; Investigation; Methodology; Project Administration; Resources; Supervision; Validation; Visualization; Roles/Writing—Original Draft; Writing—Review and Editing. V.M.: Conceptualization; Data Curation; Formal Analysis; Methodology; Software; Validation; Visualization; Roles/Writing—Original Draft; Writing—Review and Editing. J.W.: Conceptualization; Data Curation; Formal Analysis; Methodology; Validation; Visualization; Roles/Writing—Original Draft; Writing—Review and Editing. C.P.: Conceptualization; Data Curation; Formal Analysis; Funding Acquisition; Investigation; Methodology; Project Administration; Resources; Software; Supervision; Validation; Visualization; Roles/Writing—Original Draft; Writing—Review and Editing. N.B.L.: Conceptualization; Data Curation; Formal Analysis; Funding Acquisition; Investigation; Methodology; Project Administration; Resources; Software; Supervision; Validation; Visualization; Roles/Writing—Original Draft; Writing—Review and Editing. All authors have read and agreed to the published version of the manuscript.

**Funding:** This work was funded by Janssen Canada and supported by the Princess Margaret Cancer Foundation.

**Institutional Review Board Statement:** The study was conducted according to the guidelines of the Declaration of Helsinki and approved by the Institutional Review Board of the UHN-PMCC (protocol code: 22-5330 and date of approval: 9 August 2022).

**Informed Consent Statement:** Patient consent was waived by the UHN-PMCC IRB as this study was retrospective; information abstracted for the study was already collected as part of routine clinical care and contained within the patient's health record. No additional contact with participants for information was needed. All information was kept strictly confidential.

**Data Availability Statement:** The data presented in this study are not publicly available due to the privacy of individuals. The data presented in this study may be available on reasonable request from the corresponding author.

**Conflicts of Interest:** R.M., J.W., V.M. and C.P. were employees of Pentavere Research Group Inc., Toronto, ON, Canada, at the time of this study. J.L., P.B. and L.E. have no conflicts of interest to declare. A.S. has received institutional funding from Amgen, AstraZeneca, Bristol-Myers Squibb, CRISPR Therapeutics AG, Eli Lilly, Genentech, GlaxoSmithKline, Lovance, Merck, Pfizer, and Spectrum Pharmaceuticals and has consulted on the advisory boards for AstraZeneca and Genentech. G.L. has received funding for advisory boards from Takeda, AstraZeneca, Pfizer, Amgen, Bayer, Eli Lily, Merck, Novartis, Jazz, Roche, AbbVie, J&J, and BMS; has received funding for grants from Takeda, AstraZeneca, Amgen, Bayer, and Boehringer Ingelheim; and has received funding for education sessions from Takeda, AstraZeneca, Pfizer, and Bayer. F.A.S. has received institutional funding from Lilly, Pfizer, Bristol-Myers Squibb, AstraZeneca/MedImmune, and Roche Canada; honoraria from AstraZeneca, Merck Serono, Takeda, and Daiichi Sankyo; consulted for AstraZeneca and Merck Sorono; and has stocks/ownership interest in Lilly and AstraZeneca. S.I., E.A., J.E.-P. and E.M.E. are employees of Janssen Inc. and are shareholders in Johnson & Johnson. N.B.L. has received

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
