# Peer review of "Real-World Outcomes of Patients with Advanced Epidermal Growth Factor Receptor-Mutated Non-Small Cell Lung Cancer in Canada Using Data Extracted by Large Language Model-Based Artificial Intelligence"

_curroncol, doi:10.3390/curroncol31040146_

Round 1

Reviewer 1 Report

Comments and Suggestions for Authors

This is a retrospective investigation into the treatment of NSCLC correlating mutational status to treatment path and Overall Survival. This investigation utilized the new AI resources currently becoming available to search historical records and correlate extract data. Previously this has all been performed manually which is labour intensive and expensive. The technique described represents an advance in this type of investigation.

The methodology is clearly described, and all criteria are logically based. The results are presented in a clear and comprehensive fashioned although attention could be made to the visual impact of the text in graphs 2 and 3. I did find figure 1 difficult to follow and it require several re-reading of the text while viewing the figure to understand what the authors where trying to convey, although once I did understand I couldn't see how they could express it in a easier way, so I have no recommendations for this unfortunately.

The discussion clearly illustrates the problems associated with this type of investigation where personalised therapy is used for treatment. The authors do highlight the weaknesses of their approach but the benifits seen seem to outway the deficiencies. I would be happy to see such a paper published as is althought minor changes could improve the quality further

Comments on the Quality of English Language

The use of english in the document is of a very high level although the extensive use of abreviations throughout the text can make reading it difficult for the first time. 

As indicated the text in figures 2 and 3 was too small in my printout and could be improved. 

Author Response

  • Thank you for your thorough review of the manuscript, and the insightful feedback provided. We are delighted to learn that you found the presentation of methodology, results and discussion to be clear. Your endorsement for publication is very much appreciated.
  • Further, we appreciate your suggestion to improve the quality and readability of the figures, and we have updated the figures accordingly.
  • Additionally, we recognize the extensive use of abbreviations and following a thorough review, we have removed any unnecessary ones from the text

Reviewer 2 Report

Comments and Suggestions for Authors

The paper analyzes data from a large series of patients diagnosed with advanced stage non-squamous NSCLC with EGFR mutation, using AI to collect the data.

The results of the study appear widely exposed and emphasize the benefit of novel therapies in the treatment of the examined population.

The article may be considered for publication; however, some suggestions are given below.

-The authors report that out of 613 patients analyzed, 514 are affected by adenocarcinoma. Is it possible to specify what other histologies were found?

-In line number 154, the patients included in the study are generally defined as "advanced stage", however, not all of them are affected by metastatic disease. Can the authors better explain the stages included in the analysis?

Author Response

The paper analyzes data from a large series of patients diagnosed with advanced stage non-squamous NSCLC with EGFR mutation, using AI to collect the data.

The results of the study appear widely exposed and emphasize the benefit of novel therapies in the treatment of the examined population.

The article may be considered for publication; however, some suggestions are given below.

  • Thank you for your review of the manuscript. We completely agree that the results emphasize the benefit and the current unmet need of novel therapies for patients with common EGFR mutations and patients with exon 20 insertion mutations. We are thrilled you consider it for publication, and have acted upon your suggestions, as outlined below.

-The authors report that out of 613 patients analyzed, 514 are affected by adenocarcinoma. Is it possible to specify what other histologies were found?

  • Thank you for this suggestion. We have updated Table 1 to include all other histologies found within the patient cohort.

-In line number 154, the patients included in the study are generally defined as "advanced stage", however, not all of them are affected by metastatic disease. Can the authors better explain the stages included in the analysis?

    • Thank you for raising this important point. It is important to note that all patients encompassed within the cohort were advanced stage. The category "Organ level metastatic sites at diagnosis," presented in Table 1, specifically includes metastases to the bone, brain, liver, or lungs, as being the most common sites of metastases. Consequently, certain patients, despite being advanced stage, had metastases affecting body parts other than the specified organs and were therefore omitted from this particular section of the table.
    • Recognizing the need for clarity, we have revised the footnote in Table 1 to explicitly convey this information:
    • “Further, patients may have had metastases to body parts other than the bone, brain, lung and liver, which also explains why percentages may not add up to 100%”

Reviewer 3 Report

Comments and Suggestions for Authors

1.This research focused on TReal-World Outcomes of Patients with Advanced EGFR-Mutated Non-Small Cell Lung Cancer in Canada Using Data Extracted by Large language Model-Based Artificial Intelligence

, after check the pubmed, there were  not so many articles aboult this topic, so this manuscript was   very prospective and significant.

2.This manuscript foucus on clinical problems of lung cancer, with strong clinical value and importantce,very interesting research, and also met the submission topic of this journal,the results was real and the conclusion was convincing, but some places can be more perfect.

3. Atlhough target the ex20ins have good effect, but only acount for 1%, how do you treat most advanced NSCLC?

4.In patients introduction, you should emphasize all samples only have EGFR mutation, not combinded with ALK or ROS1 mutation together.

5.In 3.1 if add a flow chart maybe much more better.

6.Figure 3 dpi should much more than 300.

7.Language can be more polish.

Comments on the Quality of English Language

Nearly OK.

Author Response

1.This research focused on Real-World Outcomes of Patients with Advanced EGFR-Mutated Non-Small Cell Lung Cancer in Canada Using Data Extracted by Large language Model-Based Artificial Intelligence, after check the pubmed, there were not so many articles aboult this topic, so this manuscript was   very prospective and significant.

  • Thank you. We appreciate your thorough review and comparison to currently published work in this area. We agree that there is currently a lack of published Canadian real-world evidence on patients with advanced EGFR-mutated NSCLC, and with the treatment landscape changing, there is a need to better understand these patients and who may benefit from novel therapies.

2.This manuscript foucus on clinical problems of lung cancer, with strong clinical value and importantce,very interesting research, and also met the submission topic of this journal,the results was real and the conclusion was convincing, but some places can be more perfect.

  • Thank you, we greatly appreciate these comments and are thrilled you found the manuscript important and interesting. We also agree that this manuscript aligns well with the scope and standards of Current Oncology, but also appreciate your suggestions to improve the manuscript.
  1. Atlhough target the ex20ins have good effect, but only acount for 1%, how do you treat most advanced NSCLC?
  • As per the Canadian Cancer Society, the treatment approach for advanced non-small cell lung cancer (NSCLC) is predominantly contingent on the presence of actionable driver mutations. Should such mutations be identified, targeted therapy tailored to the specific genetic alteration are common treatment options. Noteworthy gene pathways associated with NSCLC include EGFR, ALK, BRAF, MET, ROS1, and RET, and play key roles as both causative factors and indicators for treatment.
  • In the instance of EGFR-mutated NSCLC, the location of the mutation within the exon also becomes a crucial determinant in determining an effective treatment plan. Common EGFR mutations (exon 19 deletions and exon 21 L858R point mutations) – accounting for approximately 90% of all EGFR mutations – warrant targeted therapies including erlotinib, gefitinib, and osimertinib. Presently, osimertinib is the recommended first-line therapy for advanced-stage NSCLC with common EGFR mutations in Canada.
  • In cases where no genetic mutations are identified, a comprehensive treatment approach may involve chemotherapy, immunotherapy, radiation therapy, and surgery, contingent upon the specific metastatic sites and the overall health of the patient.

4.In patients introduction, you should emphasize all samples only have EGFR mutation, not combinded with ALK or ROS1 mutation together.

  • Thank you for this comment. This study included 2,154 patients who were ≥18 years of age, with non-squamous NSCLC and seen at UHN-PMCC during the study period. This cohort was stratified based on their disease stage and EGFR mutation status at diagnosis.
  • As such, 136 patients had common EGFR mutations at diagnosis, 8 had exon 20 insertion mutations at diagnosis, 131 did not have EGFR testing at UHN and 338 were EGFR wild type. It is important to note that the EGFR wild type cohort included patients who tested negative for EGFR within 3 months of NSCLC diagnosis. Although negative for EGFR, it is possible these patients may have tested positive for another genetic mutation such as ALK or ROS1. To enhance transparency regarding the potential presence of alternative mutations in the EGFR WT cohort, the following footnote has been incorporated into
    Table 1:
  • bIncludes patients with a negative EGFR test within 3 months of NSCLC diagnosis but does not exclude the possibility of other mutations.”
  • Additionally, we have included a flow diagram of included patients in supplementary Figure 1, so it is easier to understand the patient cohorts. 

5.In 3.1 if add a flow chart maybe much more better.

  • Thank you for this suggestion. We agree that a flow chart may help readers to better understand the included patients, and the breakdown of cohorts based on EGFR mutation status. We have included the flow chart to the supplementary material Figure 1.

6.Figure 3 dpi should much more than 300.

  • Thank you for this comment. We have improved the quality of all the figures, to improve the clarity and readability.

7.Language can be more polish. 

  • Thank you, we have reviewed the manuscript for spelling, grammar and ensuring language is clear and concise, and made updates where appropriate.

Reviewer 4 Report

Comments and Suggestions for Authors

The authors reported the results of 136 cases of NSCLC with EGFR mutations extracted by a Large Language Model-Based Artificial Intelligence. It is an interesting attempt to adapt AI in clinical research. However, there are several flaws in this research.

Firstly, the authors should briefly identify the configuration of DARWEN and how the dataset was extracted in a separate paper. The concept of medical records is extensive. It is unclear how parameters such as metastatic sites, OS (Overall Survival), and TTD (Time to Treatment Discontinuation) were identified; were they determined by CT scans or physicians' records?

Secondly, the authors should validate the data extracted by DARWEN. Without this validation, the paper only offers limited novelty to our knowledge.

Author Response

The authors reported the results of 136 cases of NSCLC with EGFR mutations extracted by a Large Language Model-Based Artificial Intelligence. It is an interesting attempt to adapt AI in clinical research. However, there are several flaws in this research.

Firstly, the authors should briefly identify the configuration of DARWEN and how the dataset was extracted in a separate paper. The concept of medical records is extensive. It is unclear how parameters such as metastatic sites, OS (Overall Survival), and TTD (Time to Treatment Discontinuation) were identified; were they determined by CT scans or physicians' records?

  • Thank you for your comments. DARWENTM has been previously configured and validated against manual extraction for the same clinical features on a dataset of advanced stage lung cancer patients at UHN-PMCC which has previously been described.1
  • Models were fine-tuned and validated on the data provisioned by UHN-PMCC, to match the pre-defined clinical feature definitions and data extraction rules for this study.
  • From the data provisioned, one subset of patient data was used for fine-tuning the algorithms based on the finalized feature definitions and extraction rules.
  • The models were then run on all remaining data, which had not been part of the fine-tuning subset, to produce the final dataset which was validated against manually curated data. Accuracy, precision (positive predictive value), recall (sensitivity), and F1 (harmonic mean of precision and recall) metrics were reported.
    • Please note, validation is only appropriate for clinical features extracted using AI. Therefore, as OS and TTD are considered 'derived' features – manually calculated utilizing extracted data – AI validation is not applicable to these features.

Secondly, the authors should validate the data extracted by DARWEN. Without this validation, the paper only offers limited novelty to our knowledge.

  • We completely agree on the importance of including validation metrics, to ensure confidence in the results, and we thank the reviewer for this suggestion. As such, we have included the validation metrics for features extracted using DARWENTM in supplementary Table 1. Additionally, we have also added in the following sentence to the methods section to clarify that DARWENTM validation using UHN-PMCC lung cancer data has been previously described:
  • “DARWENTM AI has previously been validated against manual chart review for the same clinical features at UHN-PMCC, the process for which has previously been described.24

References:

  1. Gauthier, M. P.; Law, J. H.; Le, L. W.; Li, J. J. N.; Zahir, S.; Nirmalakumar, S.; Sung, M.; Pettengell, C.; Aviv, S.; Chu, R.; Sacher, A.; Liu, G.; Bradbury, P.; Shepherd, F. A.; Leighl, N. B. Automating Access to Real-World Evidence. JTO Clin Res Rep 2022, 3 (6), 100340. https://doi.org/10.1016/J.JTOCRR.2022.100340

Round 2

Reviewer 1 Report

Comments and Suggestions for Authors

just to point out line 257 of disscusion might have a mistype, you have replaced amivantamab with 11amivantamab, not sure why?? Other than that no issues, to note the figures in my opinion have been significantly improved and the refined text is much easier to read.

Reviewer 2 Report

Comments and Suggestions for Authors

I appreciated the corrections made by the authors.

Thank you for giving me the opportunity to review the paper.

Reviewer 4 Report

Comments and Suggestions for Authors

I have no additional comment on this manuscript.